# Do Determinants of Quality of Life Differ in Older People Living in the Community and Nursing Homes?

**DOI:** 10.3390/ijerph20020916

**Published:** 2023-01-04

**Authors:** Małgorzata Pigłowska, Tomasz Kostka, Agnieszka Guligowska

**Affiliations:** Department of Geriatrics, Healthy Ageing Research Centre, Medical University of Łódź, Hallera 1 Square, 90-647 Łódź, Poland

**Keywords:** nutritional status, older people, physical activity, quality of life

## Abstract

Objectives: The aim of the present study was to examine and compare the relationship between nutritional status, physical activity (PA) level, concomitant chronic diseases, and quality of life (QoL) in community-dwelling (CD) older people and nursing home (NH) residents. Material and Methods: One hundred NH residents aged 60 years and above and one hundred sex- and age-matched CD older adults were examined. The QoL was examined with the EuroQol-5D questionnaire. Nutritional status was assessed with the Mini Nutritional Assessment questionnaire (MNA), anthropometric measures, and bioimpedance analysis (BIA). The 7-Day Recall Questionnaire and the Stanford Usual Activity Questionnaire were performed to evaluate the PA energy expenditure level (PA-EE) and the health-related behaviours (PA-HRB), respectively. Results: CD subjects presented a significantly higher self-assessment in the VAS scale in comparison with NH residents (CD: 65.3 ± 19.4 vs. NH 58.2 ± 21.4; *p* < 0.05), but there were no differences within the five dimensions of QoL. In NH patients, the VAS scale was not correlated with any of the variables evaluating the nutritional status and body composition, while in the CD group correlated positively with MNA (rS = 0.36; *p* < 0.001), % of FFM (rS = 0.22; *p*< 0.05), body density (rS = 0.22; *p* < 0.05) and negatively with % of FM (rS = −0.22; *p* < 0.05). In an institutional environment, only concomitant diseases (mainly urinary incontinence) were found as independent determinants for QoL. In the community, independent determinants of QoL besides concomitant diseases (mainly ischaemic heart disease) were nutritional status or PA-HRB. Conclusions: Determinants of QoL are different depending on the living environment the older adults. Proper nutritional status and beneficial PA behaviours, are crucial for higher QoL of CD elderly, while for NH residents, the main determinants of QoL are chronic conditions.

## 1. Introduction

According to the definition proposed by the WHO, quality of life (QoL) is a subjective multidimensional construct reflecting functional status, emotional and social well-being, as well as general health [1]. In times of demographic changes connected with an increased proportion of older people worldwide, QoL has gained increased interest. Maintaining a high QoL rather than just life prolongation is one of the most important outcomes of care services for older adults, and its assessment may add a supplemental dimension in the holistic approach [2,3].

Because QoL is a crucial aim of geriatric medicine, various studies connected with its determinants have been performed. Based on the available literature, the most important and the most common factors connected with QoL of older people are, inter alia: chronic conditions, functional status, living environment, nutritional status, and physical activity (PA) [4]. However, in various studies, different confounders are included in the particular models. Moreover, the results show that some variables are specifically associated to QoL in some countries, but not in others [5].

Among the different factors that may be connected with QoL in older people, improper nutritional habits and physical inactivity seem to be the most crucial since they are modifiable and moreover, have become a major problem in the field of public health [6]. The prevalence of malnutrition in older subjects is very high and, unfortunately, very often remains unrecognized. Likewise, low PA becomes a major problem due to its influence on functional decline in advanced age [7]. The elderly population is not a homogenous one, but one that differs in various features. For example, a person’s living environment is one of the most important determinants of QoL, and the question arises of whether the nutritional status and PA are of similar importance when evaluating institutional and domestic settings.

The aim of the present study was to examine and compare the relationship between nutritional status, PA level, concomitant chronic diseases, and QoL in community-dwelling (CD) older people and nursing home (NH) residents and to find out which are independent determinants of QoL in the two environments.

## 2. Materials and Methods

### 2.1. Participants and Study Design

The conducted study was a case-control study. The participants consisted of two hundred people aged 60 years and over (150 women; 50 men) living in either domestic or institutional environments. In total, one hundred NH residents who met the inclusion criteria were enrolled to the study and then one hundred community-dwelling sex- and age-matched (±1 year of age) outpatients of the Geriatrics Clinic were subsequently included. All included participants gave written informed consent to participate in the study. The study was approved by the Bioethics Committee of the Medical University of Łódź (Project identification code: RNN/863/11/KB) and complies with the Declaration of Helsinki and Good Clinical Practice Guidelines. All tests were performed in accordance with relevant guidelines and regulations. All authors consent to the publication of this manuscript. Detailed recruitment procedure and the inclusion and exclusion criteria were described previously in our latest paper [8]. The inclusion criteria were: age (60 years and more), ability to walk (also walk using auxiliary tools), Mini Mental State Examination (MMSE) test score ≥ 11, and verbal-logical contact. The exclusion criteria were: MMSE test score < 11 points, lack of ability to walk, and the presence of palliative conditions. Additionally, pacemaker users and patients with edema were also excluded because of the requirements for the bioelectrical impedance analysis (BIA) method.

### 2.2. Data Collection

A multidimensional assessment including sociodemographic data, health status (concomitants diseases), nutritional status, PA, and QoL was performed with each participant. Moreover, the presence of palliative conditions, ability to walk, using the pacemaker and MMSE score have been checked to verify whether the patients may be included to the study.

#### 2.2.1. Sociodemographic Data

Age, gender and living environment were verified in each participant.

#### 2.2.2. Chronic Diseases

The prevalence of the following chronic diseases was assessed in the study on the basis of medical history: arterial hypertension, hypercholesterolemia, diabetes, myocardial infarction, ischaemic heart disease, chronic heart failure, stroke, osteoarthritis, osteoporosis, eye diseases, depression, chronic obstructive pulmonary disease, gastrointestinal diseases, urinary incontinence, faecal incontinence and cancer.

#### 2.2.3. Nutritional Status

Anthropometric measurements, the Mini Nutritional Assessment (MNA) questionnaire, and a body composition assessment were used to evaluate the nutritional status of examined subjects. The anthropometric tests included body mass and height, waist and calf circumferences, and calculations of body mass index (BMI) and waist-to-height ratio (WHtR) [9]. The MNA questionnaire is a validated tool that reflects the nutritional status of older people. It includes various aspects connected with the risk of malnutrition (general assessment, anthropometric measures, dietary assessment, self-assessment). The full MNA (composed of 18 questions) questionnaire was performed with a total score of 0–30. A score of 24 or more points indicates a normal nutritional status, 17–23.5 points the risk of malnutrition, and below 17 points—malnutrition [10]. A Bioimpedance analysis (BIA) was performed to assess body composition (Maltron BioScan 920-2, Maltron International Ltd., Rayleigh, UK) [11]. A whole-body analysis was performed in a supine position on the right side of the body as per the manufacturer’s instructions. The participants completed a minimum of six hours of fasting before the measurement. Fat mass (FM), fat-free mass (FFM), total body water (TBW), and body cell mass (BCM) were obtained and expressed as body mass percentage (%). Resting metabolic rate (RMR) was presented in kilocalorie/kilogram (kcal/kg), body volume in liters (L), and body density in kg/L. The extracellular water (ECW) and intracellular water (ICW) ratio were obtained. Skeletal muscle mass (SMM) (kg) [12] and skeletal muscle index (SMI) (kg/m^2^) [13] were calculated. The raw bioimpedance data: impedance (Z, Ohm), resistance (R, Ohm), reactance (Xc, Ohm), and the phase angle (PhA, degrees) was also presented.

#### 2.2.4. Physical Activity (PA)

PA was assessed with the Seven Day Recall PA Questionnaire [14] and the Stanford Usual Activity Questionnaire [15] using the standardized protocols. The Seven Day Recall PA Questionnaire is comprised of questions on the amount of sleep obtained, along with the amount of time completing light, moderate, hard, and very hard activities in the last 7 days [16]. As a result, this questionnaire estimates the average daily physical activity energy expenditure (PA-EE) over the past week (kcal·kg^−1^ day^−1^). The Stanford Questionnaire serves to assess the health-related PA behaviors of light and moderate intensity. Moderate activity was reported as six habitual activities: climbing the stairs instead of using the elevator, walking instead of driving for a short distance, parking the car further away from the destination to approach on foot, walking before or after lunch or dinner, exiting the bus/tram a stop earlier to walk the remaining distance, or performing other activities of a similar nature. The Stanford Moderate index helped classify physical activity health-related behaviors (PA-HRB). Because only five of the CD subjects and none of the NH residents reported Stanford Vigorous activity, it was not used in this paper.

#### 2.2.5. Quality of Life (QoL)

The QoL assessment of the examined subjects was performed with a popular instrument for describing the health-related quality of life (HRQoL): the EuroQol-5D questionnaire [17]. In this questionnaire, the respondents had the opportunity to report the presence of a problem in each of the five following dimensions: mobility, self-care, usual activities, pain/discomfort, and anxiety/depression, using a 1–3 scale (1 = no problems, 2 = some problems, 3 = severe problems). Additionally, the visual analogue scale (VAS) was included to record the perception of the overall health on a 0–100 scale with 0 denoting the worst imaginable health status and 100 the best imaginable health status [18].

#### 2.2.6. Statistical Analysis

To perform a statistical analysis of data, the Statistica version 13 CSS software (StatSoft, Krakow, Poland) was used. Data were verified for normality of distribution and equality of variance. To compare the nutritional status, PA and VAS scales were evaluated in both environments, and the one-way analysis of variance (ANOVA) and Mann–Whitney test were used. A Chi^2^ test was used for comparisons between categorical variables. EuroQol 5D dimensions data were dichotomised (no problem vs. any problem). The results of the quantitative variables are presented as mean ± SD (standard deviation) for data with a normal distribution, and additionally as median and quartiles for data without normal distribution. The correlations were assessed with Spearman correlation coefficients. The association between the prevalence of particular diseases and QoL was verified using a Chi^2^ test (for the EuroQol dimensions) and the Mann–Whitney test (for the VAS scale). Variables statistically significant in bivariate analysis were included in a multifactorial model. A logarithmic transformation was performed to achieve a normal distribution of quantitative data, which was used in the multifactorial analysis. Stepwise logistic regression was used to assess which independent variables were independent predictors for particular QoL dimensions in both environments. A general linear model was performed to find independent factors connected with the VAS scale in both groups. The limit of significance was assumed to be a *p*-value of 0.05 or less for all analyses.

## 3. Results

Table 1 presents a comparison of age, MMSE and the prevalence of chronic diseases in the two environments.

The age of the two examined groups was not statistically different. Each group consisted of approximately 75% women and 25% men. Chronic diseases of the study subjects are shown in Table 1. As we presented in our previous article, nutritional status varies between living environments [8]. The CD older people were characterized with better nutritional status in comparison with NH residents. For example, they obtained significantly higher results in MNA questionnaire (73% presented proper nutritional level compared to 23% in NH; 2% vs. 14% subjects indicated malnutrition in the community and NH, respectively). They also presented significantly higher anthropometric (body mass, calf circumference, and BMI), and several different body composition indices.

The QoL and PA in the NH and CD groups are shown in Table 2. CD subjects presented a significantly higher self-assessment in the VAS scale. However, the assessment of QoL did not differ between the groups in any of the five examined dimensions. PA of CD older people was significantly higher in comparison with NH residents measured as PA-EE as well as PA-HRB.

Table 3 presents correlations between selected indicators of nutritional status, VAS scale, and PA in both groups. A higher level of PA was connected with better nutritional status in both environments. PA-HRB was related to several nutritional parameters in both environments. PA-EE in the CD group was associated only with MNA, while in NH residents, PA-EE was associated with many components of nutritional status. VAS scale was related to some of the nutritional indices and PA-HRB in CD subjects but not in NH residents.

The relationships of the EuroQol-5D test dimensions to age, nutritional parameters, and PA in NH residents and CD subjects are shown in Appendix A (Appendix A, respectively).

The relationships of the EuroQol-5D test dimensions to age, nutritional parameters, and PA were different in both groups. In a domestic environment, several nutritional parameters, as well as PA, were found to be related to QoL dimensions. MNA was connected with all domains of QoL. Moreover, BMI, WHtR, FFM, FM, body density, RMR, TBW, ECW/ICW, BCM, SMI, PhA and PA-HRB were related to mobility; PhA, PA-HRB to self-care; ECW/ICW and PhA to Usual activity; FFM, FM, body density, TBW, BCM, to pain/discomfort; BMI, WHtR, FFM, FM, body density, RMR, TBW, BCM, PA-EE and PA-HRB with anxiety/depression. In NH residents, only a few of these relationships were present: MNA, PhA and PA-HRB were connected with mobility; MNA with pain/discomfort; and MNA, FFM, FM, body density, BCM, PA-EE and PA-HRB with anxiety/depression.

There was an association between the prevalence of particular diseases and QoL found in both environments. In NH residents, the prevalence of hypertension was connected with deteriorated mobility; hypercholesterolemia with mobility, pain/discomfort, and anxiety/depression; ischaemic heart disease with usual activities; heart failure with self-care; myocardial infarction with mobility; stroke with self-care; osteoporosis with mobility and anxiety/depression; osteoarthritis with mobility, self-care, pain/discomfort, and anxiety/depression; gastrointestinal diseases with mobility; eye diseases with pain; urinary incontinence with VAS, mobility, self-care, and anxiety/depression.

In the CD group, the prevalence of hypertension was connected with worse VAS; ischaemic heart disease with worse VAS, mobility, usual activities, pain/discomfort, and anxiety/depression; myocardial infarction with worse mobility and usual activities; heart failure with worse VAS, pain/discomfort and anxiety/depression; stroke with worse VAS and usual activities; osteoarthritis with worse mobility, self-care and pain/discomfort; osteoporosis with worse VAS, mobility, and usual activities; the presence of depression with worse VAS, mobility and anxiety/depression; urinary incontinence with worse VAS, mobility, and self-care.

Table 4 presents the independent determinants of the VAS scale and the particular dimensions of the EuroQol-5D test in the two environments selected in multivariate analyses. It revealed that, in the community, the nutritional indices and PA-HRB were found, besides concomitant diseases, as independent determinants of QoL. On the other hand, in an institutional environment, only concomitant diseases were found as independent determinants for all particular elements of QoL. The chronic disease that was most often presented as an independent factor in this type of environment was urinary incontinence while ischaemic heart disease was the most prevalent in CD subjects.

## 4. Discussion

The present study extends our previously reported data showing that the determinants of QoL of older adults are different in the community and institution [19]. Older individuals in the CD group are characterized with higher nutritional status, PA and QoL. The relationship between nutritional status and PA is visible and even higher in the NH group. Nevertheless, the impact on QoL seems to be different.

Most of the existing studies show that QoL is usually better in a domestic environment in comparison with an institution, and that institutionalization significantly predicts negative changes over time in QoL [20,21,22]. Our results indicate that the perception of the overall health in the VAS scale is significantly better in subjects living in the community. In the present study, we analysed the nutritional status and PA level (which are crucial factors for maintaining a high QoL) of older people living in different environments [20,23,24,25]. Our result supported the finding that the nutritional state of institutionalized subjects is often worse than those living in the community [20,26]. It has also been observed that institutionalized older adults often have a lower level of PA than CD older adults [21,27,28,29], and that the PA level is found to decrease in over 30% of study participants during institutionalization [30]. In our study, older individuals who lived in NH were less active and showed significantly more sedentary behaviors in comparison with CD subjects. The differences were connected with both PA-EE and PA-HRB which indicates that less time is spent on leisure time PA and everyday chores, and also a lower number of health-related behaviors during everyday activities.

The multifactorial analyses in our study showed that independent nutritional determinants of QoL in the NH group were concomitant diseases. In the community group, these diseases were also found to be important determinants of QoL; however, the nutritional status and PA-HRB showed crucial contributions in determining QoL in this group. Similar results were obtained in another study performed almost 20 years ago, but without bioimpedance data [19]. It was found that primary predictors of life quality in the CD group were overweight/obesity and PA, while in institutionalized subjects these factors were not so important and concomitant diseases were dominant determinants of QoL. Available data show that gender has been found as an independent determinant with a positive impact on the QoL assessment in males [31], which is consistent with our results in pain/discomfort and anxiety/depression domains in the CD group.

The relationship between nutritional status and QoL has been observed in many studies [20,23,31,32,33,34,35]. Our study revealed the differences in the relationship between QoL and nutritional status in the examined groups of older people. Multifactorial analysis indicated that nutritional status had an independent contribution to the prediction of QoL in CD older subjects. The nutritional factor which the most often determined life quality was MNA. The MNA includes the assessment of different spheres of life and, therefore, it is suggested to be an effective indicator connected not only with nutritional status but also with functional state, muscle strength, or mortality [36]. Among raw bioelectrical parameters obtained in BIA analyses neither R (connected with body water) nor Xc (connected with BCM) was independent determinants of QoL, whereas PhA (which is connected simultaneously with the metabolically active part of FFM and body fluids, and also assesses functional reserves and correlates with muscle strength [37]), was a significant determinant for self-care. The first possible cause of these differences in the two environments is generally poorer nutritional status in NH subjects. Second, we should consider the additional indirect influence of nutritional status and PA on QoL. For example, nutritional status is connected [38] and may partially modulate functional state which influences QoL. According to our previous paper, nutritional status was a crucial determinant of functional state for older people living in the community but less so for NH residents [8].

Being active is one of the most important factors for maintaining QoL in older people [4,25]. Interpreting the differences in the relationship between PA and QoL in both environments, we should take into consideration that it might be indirect and mediated by other variables. An indirect effect of PA in older people was previously observed by other authors—PA influenced overall QoL through socio-psychological variables. Moreover, the effect of PA on QoL might be influenced by perceived social support, which is an important aspect of QoL [4,39]. Therefore, the fact that the PA of NH residents was lower in comparison with CD subjects is very important, but it should be also be taken into account that NH residents may experience a decrease in autonomy, feel a sense of homelessness, or experience social isolation [40]. The differences in the assessment of QoL dependent on social support were also visible in the results obtained by Chruściel et al. [41]—older people who lived with their families assessed the majority of the QoL domains significantly better than those who lived alone. According to Hajek et al. [42], decreased social support was associated with higher possibility of developing problems in most of EuroQol-5D dimensions within individuals over time. Family routine activities and social participation may prevent functional deterioration and affects the QoL of the elderly [22]. Another aspect of the examined PA of the participants of our study was that most of the PA and QoL correlations in both environments were present between the QoL and PA-HRB, while PA-EE was connected only with anxiety/depression. Moreover, PA-HRB was the independent determinant of mobility in the CD group. It shows that the health-related behaviors seem to be more important for the QoL than the time spent on usual PA (activity assessed in the Stanford Usual Activity Questionnaire). We may speculate that it may be connected with higher awareness of the positive influence of these choices on health. In the other study, more frequent PA-HRB was found in people with higher education levels [29]. Taking more frequent activities just “for health” may be also partially connected with social support that was described above—the positive effect of the company may be present as higher motivation in some activities.

Different living environment causes differences in the physical, social and psychological aspects of elderly people’s wellbeing, and it may affect their health status. The prevalence of chronic conditions may influence different life aspects which indirectly significantly worsen the QoL of older adults [43]. In our study, concomitant diseases were found to be independent determinants of all aspects of QoL in NH residents, but they were also present in subjects from the community. Chronic conditions influence older people’s functional status and people suffering from multimorbidity very often present lower PA than older people with a lower number of chronic diseases [29]. On the other hand, increasing the PA level is crucial in most chronic conditions. It is worth mentioning that the QoL may be affected by chronic conditions; however, probably not only the presence of the diseases may be an important issue for QoL, but it also depends on adaptive forces of coping with the diseases [31]. Klompstra et al. [44] found that a higher symptom burden was negatively connected with QoL in older CD people with multimorbidity. The assessment of QoL may also allow researchers to evaluate the effects of medical interventions. In our study, urinary incontinence and cardiac diseases were important determinants of QoL in the two environments. Patients with cardiac diseases often suffer from symptoms such as fatigue, shortness of breath, nausea, and oedema [45]. The relationships between cardiovascular diseases, including ischaemic heart disease, and QoL has been observed and assessing how this disease impacts the patient’s well-being is important [46]. Urinary incontinence may affect the physical, psychological, and social aspects of older people’s life [47] and is associated with a poor QoL [48]. In NH residents, it was the most important chronic condition affecting QoL. Several mechanisms connected with the influence of urinary incontinence on QoL have been described [48]. Among them is the fact that seniors who suffer from urinary incontinence more often experience also other health problems and, therefore, the additional factors connected with them may occur (for example, diuretic treatment) and exacerbate urinary incontinence [49]. The presence of urinary incontinence may also lead to decreasing PA levels. Moreover, incontinence-associated dermatitis may arise and deteriorate QoL [48,50]. This problem is also associated with depression, stress and self-esteem [51]. The role of relatives, caregivers, and their support may be important to overcome the shame feeling, to accept the presence of this problem and thus to start intervention [47]. Lack of social support may be partially connected with the importance of this problem for several dimensions of QoL in older NH residents.

This study includes some limitations. The cross-sectional design and relatively small sample must be acknowledged as major limitations of our study and restricts the possibility of drawing any conclusions on causality. The assessment of PA level was self-reported by the participants, which may cause desirability bias. Moreover, our sample is non-representative and may not be generalizable to the general older adult population since the participants were recruited in the Geriatric Clinic, not in the primary care setting.

## 5. Conclusions

Our findings implicate the need for a different approach when considering elderly individuals who live in communities verses nursing homes. Besides chronic illnesses, maintaining proper nutritional status and beneficial behaviours connected with PA seems to be crucial for QoL of CD older subjects, while focusing on the existing chronic conditions is the first step necessary to improve QoL of NH residents.

## Figures and Tables

**Table 1 ijerph-20-00916-t001:** A comparison of age, MMSE and the prevalence of chronic diseases in the two environments.

Variables	NH N = 100	CD N = 100	*p*-Value
Age (years)	74.6 ± 9.74	74.9 ± 8.50	ns
Women (%)	75	75	ns
MMSE	23.7 ± 4.69 24.0 (22.0; 27)	26.8 ± 3.84 28.0 (25.0; 30.0)	<0.001
Arterial hypertension (n)	67	63	ns
Hipercholesterolemia (n)	38	63	<0.001
Diabetes (n)	25	24	ns
Myocardial infarction (n)	11	9	ns
Ischaemic heart disease (n)	56	36	0.005
Chronic heart failure (n)	56	41	0.034
Stroke (n)	20	13	ns
Osteoarthritis (n)	56	57	ns
Osteoporosis (n)	25	38	0.048
Eye disease (%)	52	56	ns
Depression (n)	32	24	ns
Chronic obstructive pulmonary disease (n)	20	13	ns
Gastrointestinal diseases (n)	27	40	0.051
Urinary incontinence (n)	40	37	ns
Faecal incontinence (n)	7	8	ns
Cancer (n)	11	11	ns

MMSE: Mini Mental State Examination; NH: nursing home group; CD: community-dwelling group; ns: not significant.

**Table 2 ijerph-20-00916-t002:** A comparison of QoL-5D, VAS scale and PA in the NH and CD groups.

EuroQol-5D	NH N = 100			CD N = 100			*p*-Value
	A	B	C	A	B	C	
Mobility	54	43	3	48	50	2	ns
Self-care	65	34	1	76	21	3	ns
Usual activities	53	45	2	52	43	5	ns
Pain/discomfort	19	74	7	16	76	8	ns
Anxiety/depression	31	64	5	21	75	4	ns
VAS scale	58.2 ± 21.4 60.0 (50.0; 70.0)			65.3 ± 19.4 62.5 (50.0; 80.0)			0.041
PA-EE (kcal/kg/day)	35.09 ± 3.71 34.20 (32.6; 36.9)			38.84 ± 6.34 37.0 (35.0; 41.6)			<0.001
PA-HRB	1.05 ± 1.04 1.0 (0.0; 2.0)			1.52 ± 1.28 1.0 (1.0; 2.0)			<0.001

For statistical analysis Euroqol 5D dimensions data were dichotomised (no problem vs. any problem). VAS scale: visual analogue scale; PA-EE: physical activity energy expenditure; PA-HRB: physical activity health related behaviours; NH: nursing home group; CD: community-dwelling group; ns: not significant.

**Table 3 ijerph-20-00916-t003:** Statistically significant correlations between VAS scale/PA and age/selected indicators of nutritional status in both groups.

NH				CD			
N = 100	VAS Scale	PA-EE	PA-HRB	N = 100	VAS Scale	PA-EE	PA-HRB
Age		−0.22 *	−0.22 *	Age		−0.28 **	−0.31 **
MNA		0.21 *	0.26 **	MNA	0.36 ***	0.39 ***	0.40 ***
BMI		−0.22 *		BMI			
WHtR		−0.28 **		WHtR			−0.21 *
Waist (cm)				Waist (cm)			
Calf (cm)		−0.20 *		Calf (cm)			
FFM (%)		0.30 **	0.22 *	FFM (%)	0.22 *		0.25 *
FM (%)		−0.30**	−0.22 *	FM (%)	−0.22 *		−0.25 *
Body volume (L)				Body volume (L)			
Body density		0.30 **	0.22 *	Body density	0.22 *		0.25 *
RMR (kcal/kg)		0.28 **	0.21 *	RMR (kcal/kg)			0.22 *
TBW (%)		0.22 *		TBW (%)			
ECW/ICW			−0.20 *	ECW/ICW			−0.21 *
BCM (%)		0.23 *	0.22 *	BCM (%)			
SMI (kg/m^2^)				SMI (kg/m^2^)			
Z (Ω)				Z (Ω)			
R (Ω)				R (Ω)			
Xc (Ω)		0.22 *	0.40 ***	Xc (Ω)			0.21 *
PhA		0.22 *	0.46 ***	PhA			0.31 **
PA-EE		-		PA-EE		-	
PA-HRB		0.46 ***	-	PA-HRB	0.30 **		-

* *p* < 0.05; ** *p* < 0.01; *** *p* < 0.001; MNA: Mini Nutritional Assessment; BMI: body mass index; WHtR: waist-to-height ratio; FM: fat mass; FFM: fat-free mass; RMR: resting metabolic rate; TBW: total body water; ECW/ICW: extracellular water and intracellular water ratio; BCM: body cell mass; SMI: skeletal muscle index; Z: impedance; R: resistance; Xc: reactance; PhA: phase angle; Ω: Ohm; VAS scale: visual analogue scale; PA-EE: physical activity weekly energy expenditure; PA-HRB: physical activity health related behaviours; NH: nursing home group; CD: community-dwelling group.

**Table 4 ijerph-20-00916-t004:** Independent determinants of particular elements of QoL in both environments selected in multivariate analyses.

Dependent Variable	Significant Independent Variables	
	NH	CD
VAS scale	Urinary incontinence (−) **	MNA (+) ***
Mobility (no problems)	Hypercholesterolemia (−) * Myocardial infarction (−) ** Osteoporosis (−) ***	Depression (−) *** Urinary incontinence (−) ** TBW (%) (+) ** PA-HRB (+) **
Self-care (no problems)	Stroke (−) ** Urinary incontinence (−) **	MNA (+) ** PhA (+) **
Usual activities (no problems)	Ischaemic heart disease (−) *	MNA (+) * Ischaemic heart disease (−) **
Pain/Discomfort (no problems)	Hypercholesterolemia (−) ** Osteoarthritis (−) ***	Sex (+while man) ** Ischaemic heart disease (−) ***
Anxiety/Depression (no problems)	Urinary incontinience (−) **	Sex (+while man) ** Ischaemic heart disease (−) *** BMI (−) *

(+)—positive predictor for the QoL; (−)—negative predictor for the QoL. * *p* < 0.05, ** *p* < 0.01, *** *p* < 0.001. VAS scale: visual analogue scale; MNA: Mini Nutritional Assessment; TBW: total body water; %: percentage; BMI: body mass index; PhA: phase angle; Ω: Ohm; PA-HRB: physical activity health related behaviours; NH: nursing home group; CD: community-dwelling group.

## Data Availability

The data from nursing home residents may be potentially identifiable. Therefore, the data are not publicly available. The datasets used and/or analyzed during the current study are available from the corresponding author on reasonable request.

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
