# Peer review of "Do Determinants of Quality of Life Differ in Older People Living in the Community and Nursing Homes?"

_ijerph, 2023, doi:10.3390/ijerph20020916_

Round 1

Reviewer 1 Report

Thank you very much for the opportunity to review this manuscript. The topic is interesting. However, the manuscript needs significant reorganization to organize the information and make it easier for the reader to familiarize themselves with its contents.

summary

Please structure the abstract: introduction, purpose, material and method, results, discussion, conclusions

Please think about the purpose of your work. You are studying more determinants of QOL, not just physical activity and nutritional status. The tools should indicate the sociodemographic and health variables, such as chronic diseases, that were collected. Especially since the first sentence of the results concerns diseases, which is not at all clear from the tools mentioned.

Introduction

Note as above regarding the formulation of the purpose of the work.

Material and methods:

There is a big mess in this very important part. Please adopt the structure of the STROB guidelines to describe this part in detail and accurately. Please separate sections. Please list the tools so that they are legible and transparent. Writing this part in compact text means that many elements are lost, and it is very tiring for the reader. If a questionnaire has not been analyzed in your work, please delete it

It is incomprehensible to accept MMES ³ 11. Whether dementia at this level allowed to reliably answer such a number of questionnaires. Please explain. This should probably be included in the work constraints as well.

Results

For the clarity of the description of the results, please link the specific description of the results to the specific table. The results described in one string are very illegible.

In Table 1, please show a comparison of the group in terms of MMES and other variables not included here. Please adapt the title of the table to its content.

Tab 6 Please specify the variables in the table. While it is understandable that the occurrence of a stroke will negatively affect self-care, it is not so obvious whether an increase or decrease in individual parameters of body composition. Moreover, why were these parameters chosen, what about the others from tables 5 and 6?

Discussion

Work restrictions should be extended a non-representative and relatively small sample. Cognition and reliability issues to consider.

Moreover, a very large number of data from tables 5 and 6 seems to be little used in some discussions and conclusions.

Author Response

Dear Reviewer,

We appreciate the time and effort that you dedicated to provide feedback on our paper and we are grateful for all the constructive and valuable comments that allow us to improve our manuscript.

According to your suggestions, the Abstract has been structured and divided into particular parts.

Objectives: The aim of the present study was to examine and compare the relationship between nutritional status, physical activity (PA) level, concomitant chronic diseases, and quality of life (QoL) in community-dwelling (CD) older people and nursing home (NH) residents.

Material and methods: One hundred NH residents aged 60 years and above and one hundred sex- and age-matched CD older adults were examined. The QoL was examined with the EuroQol-5D questionnaire. Nutritional status was assessed with the Mini Nutritional Assessment questionnaire (MNA), anthropometric measures, and bioimpedance analysis (BIA). The 7-Day Recall Questionnaire and the Stanford Usual Activity Questionnaire were performed to evaluate the PA energy expenditure level (PA-EE) and the health-related behaviours (PA-HRB), respectively.

Results: CD subjects presented a significantly higher self-assessment in the VAS scale in comparison with NH residents (CD: 65.3 ± 19.4 vs. NH 58.2 ± 21.4; p<0.05) but there were no differences within the five dimensions of QoL. In NH patients, the VAS scale was not correlated with any of the variables evaluating the nutritional status and body composition, while in the CD group correlated positively with MNA (rS=0.36; p<0.001), % of FFM (rS=0.22; p< 0.05), body density (rS=0.22; p<0.05) and negatively with % of FM (rS= -0.22; p<0.05). In an institutional environment, only concomitant diseases (mainly urinary incontinence) were found as independent determinants for QoL. In the community, independent determinants of QoL besides concomitant diseases (mainly ischaemic heart disease) were nutritional status or PA-HRB.

Conclusion: Determinants of QoL are different depending on the living environment of older adults. Proper nutritional status and beneficial PA behaviours, are crucial for higher QoL of CD elderly, while for NH residents the main determinants of QoL are chronic conditions.

Introduction 

Note as above regarding the formulation of the purpose of the work. 

According to the reviewer's suggestion, the purpose of the work was modified:

The aim of the present study was to examine and compare the relationship between nutritional status, PA level, concomitant chronic diseases, and QoL in community-dwelling (CD) older people and nursing home (NH) residents and to find out which are independent determinants of QoL in the two environments.

Material and methods: 

There is a big mess in this very important part. Please adopt the structure of the STROB guidelines to describe this part in detail and accurately. Please separate sections. Please list the tools so that they are legible and transparent. Writing this part in the compact text means that many elements are lost, and it is very tiring for the reader. If a questionnaire has not been analyzed in your work, please delete it 

It is incomprehensible to accept MMES ³ 11. Whether dementia at this level allowed to reliably answer such a number of questionnaires. Please explain. This should probably be included in the work constraints as well.

We thank the reviewer for his suggestions and for pointing to the STROB guidelines as a model. The tools used in the manuscript have been clarified and described. In the Methodology, particular sections have been divided. MMSE data have been provided in the Results section of the study. Only seven subjects had MMSE in the range of 11-18 points, and there were able to perform all the tests required in the study.

  1. Materials and methods

2.1. Participants and study design

The conducted study was a case-control study. The participants consisted of two hundred people aged 60 years and over (150 women; 50 men) living in either domestic or institutional environments. In total, one hundred NH residents who met the inclusion criteria were enrolled to the study and then, one hundred community-dwelling sex- and age-matched (±1 year of age) outpatients of the Geriatrics Clinic were subsequently included. All included participants gave written informed consent to participate in the study. The study was approved by the Bioethics Committee of the Medical University of Łódź (Project identification code: RNN/863/11/KB) and complies with the Declaration of Helsinki and Good Clinical Practice Guidelines. All tests were performed in accordance with relevant guidelines and regulations. All authors consent to the publication of this manuscript. Detailed recruitment procedure and the inclusion and exclusion criteria were described previously in our latest paper [8]. The inclusion criteria were: age (60 years and more), ability to walk (also walk using auxiliary tools), Mini Mental State Examination (MMSE) test score ≥11, and verbal-logical contact. The exclusion criteria were: MMSE test score<11 points, lack of ability to walk, and presence of palliative conditions. Additionally, pacemaker users and patients with edema were also excluded because of the requirements for the bioelectrical impedance analysis (BIA) method.

2.2. Data collection

A multidimensional assessment including sociodemographic data, health status (concomitants diseases), nutritional status, PA, and QoL was performed with each participant. Moreover, the presence of palliative conditions, ability to walk, using the pacemaker and MMSE score have been checked to verify whether the patients may be included to the study.

 Sociodemographic data

Age, gender and living environment were verified in each participant.

 Chronic diseases

The prevalence of the following chronic diseases was assessed in the study on the basis of medical history: arterial hypertension, hypercholesterolemia, diabetes, myocardial infarction, ischaemic heart disease, chronic heart failure, stroke, osteoarthritis, osteoporosis, eye diseases, depression, chronic obstructive pulmonary disease, gastrointestinal diseases, urinary incontinence, faecal incontinence, cancer.

Nutritional status

Anthropometric measurements, the Mini Nutritional Assessment (MNA) questionnaire , and a body composition assessment were used to evaluate the nutritional status of examined subjects. The anthropometric tests included body mass and height, waist and calf circumferences, and calculations of body mass index (BMI) and waist‐to‐height ratio (WHtR) [9]. The MNA questionnaire is a validated tool that reflects the nutritional status of older people. It includes various aspects connected with the risk of malnutrition (general assessment, anthropometric measures, dietary assessment, self-assessment). The full MNA (composed of 18 questions) questionnaire was performed with a total score of 0-30. A score of 24 or more points indicates a normal nutritional status, 17 - 23.5 points the risk of malnutrition, and below 17 points - malnutrition [10]. A Bioimpedance analysis (BIA) was performed to assess body composition (Maltron BioScan 920-2, Maltron International Ltd) [11]. A whole-body analysis was performed in a supine position on the right side of the body as per the manufacturer’s instructions. The participants completed a minimum of six hours of fasting before the measurement. Fat mass (FM), fat-free mass (FFM), total body water (TBW), and body cell mass (BCM) were obtained and expressed as body mass percentage (%). Resting metabolic rate (RMR) was presented in kilocalorie/kilogram (kcal/kg), body volume in liters (L), and body density in kg/L. The extracellular water (ECW) and intracellular water (ICW) ratio were obtained. Skeletal muscle mass (SMM) (kg) [12] and skeletal muscle index (SMI) (kg/m2) [13] were calculated. The raw bioimpedance data: impedance (Z, Ohm), resistance (R, Ohm), reactance (Xc, Ohm) and, the phase angle (PhA, degrees) were also presented.

Physical activity (PA)

PA was assessed with the Seven Day Recall PA Questionnaire [14] and the Stanford Usual Activity Questionnaire [15] using the standardized protocols. The Seven Day Recall PA Questionnaire is comprised of questions on the amount of sleep obtained, along with the amount of time completing light, moderate, hard, and very hard activities in the last 7 days [16]. As a result, this questionnaire estimates the average daily physical activity energy expenditure (PA-EE) over the past week (kcal·kg-1·day-1). The Stanford Questionnaire serves to assess the health-related PA behaviors of light and moderate intensity. Moderate activity was reported as six habitual activities: climbing the stairs instead of using the elevator, walking instead of driving for a short distance, parking the car further away from the destination to approach on foot, walking before or after lunch or dinner, exiting the bus/tram a stop earlier to walk the remaining distance, or performing other activities of a similar nature. The Stanford Moderate index helped classify physical activity health-related behaviors (PA-HRB). Because only 5 of the CD subjects and none of the NH residents reported Stanford Vigorous activity, it was not used in this paper.

 Quality of life (QoL)

The QoL assessment of the examined subjects was performed with a popular instrument for describing the health-related quality of life (HRQoL): the EuroQol-5D questionnaire [17]. In this questionnaire the respondents had the opportunity to report the presence of a problem in each of the five following dimensions: mobility, self-care, usual activities, pain/discomfort, and anxiety/depression, using a 1 - 3 scale (1 = no problems, 2 = some problems, 3 = severe problems). Additionally, the visual analogue scale (VAS) was included to record the perception of the overall health on a 0-100 scale with 0 denoting the worst imaginable health status and 100 the best imaginable health status [18].

Statistical analysis

To perform a statistical analysis of data, the Statistica version 13 CSS software (StatSoft, Krakow, Poland) was used. Data were verified for normality of distribution and equality of variance. To compare the nutritional status, PA and VAS scales were evaluated in both environments, and the one-way analysis of variance (ANOVA) and Mann-Whitney test were used. A Chi2 test was used for comparisons between categorical variables. EuroQol 5D dimensions data were dichotomised (no problem vs. any problem). The results of the quantitative variables are presented as mean ± SD (standard deviation) for data with a normal distribution, and additionally as median and quartiles for data without normal distribution. The correlations were assessed with Spearman correlation coefficients. The association between the prevalence of particular diseases and QoL was verified using a Chi2 test (for the EuroQol dimensions) and the Mann-Whitney test (for the VAS scale). Variables statistically significant in bivariate analysis were included in a multifactorial model. A logarithmic transformation was performed to achieve a normal distribution of quantitative data, which was used in the multifactorial analysis. Stepwise logistic regression was used to assess which independent variables were independent predictors for particular QoL dimensions in both environments. A general linear model was performed to find independent factors connected with the VAS scale in both groups. The limit of significance was assumed to be a p-value of 0.05 or less for all analyses.

Results 

For the clarity of the description of the results, please link the specific description of the results to the specific table. The results described in one string are very illegible. 

In Table 1, please show a comparison of the group in terms of MMES and other variables not included here. Please adapt the title of the table to its content. 

The comparison of MMSE scores in the two environments has been added in Table 1, we also change the title of the table.

The nutritional status parameters in the two environments have been presented in our previous paper and cited in the manuscript: Pigłowska M., Guligowska A., and Kostka T., Nutritional Status Plays More Important Role in Determining Functional State in Older People Living in the Community than in Nursing Home Residents. Nutrients, 2020. 12(7).

The table numbers have been linked to the relevant parts of the text.

Table 1. A comparison of age, MMSE, and the prevalence of chronic diseases in the two environments.

Variables

NH

N=100

CD

N=100

p-value

Age (years)

74.6 ± 9.74

74.9 ± 8.50

ns

Women (%)

75

75

ns

MMSE

23.7 ± 4.69

24.0 (22.0; 27)

26.8 ± 3.84

28.0 (25.0; 30.0)

<0.001

Arterial hypertension (n)

67

63

ns

Hipercholesterolemia (n)

38

63

<0.001

Diabetes (n)

25

24

ns

Myocardial infarction (n)

11

9

ns

Ischaemic heart disease (n)

56

36

0.005

Chronic heart failure (n)

56

41

0.034

Stroke (n)

20

13

ns

Osteoarthritis (n)

56

57

ns

Osteoporosis (n)

25

38

0.048

Eye disease (%)

52

56

ns

Depression (n)

32

24

ns

Chronic obstructive pulmonary disease (n)

20

13

ns

Gastrointestinal diseases (n)

27

40

0.051

Urinary incontinence (n)

40

37

ns

Faecal incontinence (n)

7

8

ns

Cancer (n)

11

11

ns

MMSE: Mini Mental State Examination; NH: nursing home group; CD: community - dwelling group.

Tab 6 Please specify the variables in the table. While it is understandable that the occurrence of a stroke will negatively affect self-care, it is not so obvious whether an increase or decrease in individual parameters of body composition. Moreover, why were these parameters chosen, what about the others from tables 5 and 6? 

Thank you for this valuable comment. The possible explanations of affecting QoL by nutritional status have been included in the Discussion. The additional indirect influence of nutritional status on QoL seems to be important - nutritional status may modulate functional status which influences QoL (also self-care). Moreover, in our previous paper, nutritional status was a crucial determinant of functional state for older people living in the community but less so for NH residents. The parameters chosen in Table 4 (previously 6) (selected in the multivariate model) were those variables statistically significant in bivariate analysis, p≤0.05.

Variables statistically significant in bivariate analysis were included in a multifactorial model. A logarithmic transformation was performed to achieve a normal distribution of quantitative data, which was used in the multifactorial analysis. Stepwise logistic regression was used to assess which independent variables were independent predictors for particular QoL dimensions in both environments. A general linear model was performed to find independent factors connected with the VAS scale in both groups.

Discussion 

Work restrictions should be extended a non-representative and relatively small sample. Cognition and reliability issues to consider. 

Work restrictions have been extended to a non-representative and relatively small sample.

This study includes some limitations. The cross-sectional design and relatively small sample must be acknowledged as major limitations of our study and restricts the possibility of drawing any conclusions on causality. The assessment of PA level was self-reported by the participants what may cause desirability bias. Moreover, our sample is non-representative and may not be generalizable to the general older adult population since the participants were recruited in the Geriatric Clinic, not in the primary care setting.

Moreover, a very large number of data from tables 5 and 6 seems to be little used in some discussions and conclusions. 

These tables have been provided as supplementary materials (suppl. Tables 1 and 2). The main results have been shown in an additional paragraph. Those variables statistically significant in bivariate analyses (p≤0.05) were used in the multivariate model and further discussed.

The relationships of the EuroQol-5D test dimensions to age, nutritional parameters, and PA in NH residents and CD subjects are shown in supplementary tables (Supplementary Table 1 and Supplementary Table 2, respectively).

The relationships of the EuroQol-5D test dimensions to age, nutritional parameters, and PA were different in both groups. In a domestic environment, several nutritional parameters, as well as PA, were found to be related to QoL dimensions. MNA was connected with all domains of QoL. Moreover, BMI, WHtR, FFM, FM, body density, RMR, TBW, ECW/ICW, BCM, SMI, PhA, PA-HRB were related to mobility; PhA, PA-HRB to self-care; ECW/ICW and PhA to Usual activity; FFM, FM, body density, TBW, BCM, to pain/discomfort; BMI, WHtR, FFM, FM, body density, RMR, TBW, BCM, PA-EE and PA-HRB with anxiety/depression. In NH residents, only a few of these relationships were present: MNA, PhA and PA-HRB were connected with mobility; MNA with pain/discomfort; and MNA, FFM, FM, body density, BCM, PA-EE and PA-HRB with anxiety/depression.

There was an association between the prevalence of particular diseases and QoL found in both environments. In NH residents, the prevalence of hypertension was connected with deteriorated mobility; hypercholesterolemia with mobility, pain/discomfort, and anxiety/depression; ischaemic heart disease with usual activities; heart failure with self-care; myocardial infarction with mobility; stroke with self-care; osteoporosis with mobility and anxiety/depression; osteoarthritis with mobility, self-care, pain/discomfort, and anxiety/depression; gastrointestinal diseases with mobility; eye diseases with pain; urinary incontinence with VAS, mobility, self-care, and anxiety/depression.

In the CD group the prevalence of hypertension was connected with worse VAS; ischaemic heart disease with worse VAS, mobility, usual activities, pain/discomfort, and anxiety/depression; myocardial infarction with worse mobility and usual activities; heart failure with worse VAS, pain/discomfort and anxiety/depression; stroke with worse VAS and usual activities; osteoarthritis with worse mobility, self-care and pain/discomfort; osteoporosis with worse VAS, mobility, and usual activities; the presence of depression with worse VAS, mobility and anxiety/depression; urinary incontinence with worse VAS, mobility, and self-care.

Reviewer 2 Report

The paper presents a novel and interesting approach to the important issue of factors determining a quality of life (QoL)in older adults living in the various environment. The two compared study group seem to be quite representative, although the results would be surely more convincing if the study could be continued  with the enlarged populations. The  performed analyses lead to the practically valuable conclusions that can improve the preventive actions undertaken in order to enhance the quality of life in elderly persons.  

The Abstract would become more interesting for readers  when supplied with some, the most significant  number results of the study.

The Introduction emphasizes the apt opinion that a high QoL in elderly population is more important than just life prolongation. Text included  between the lines 42 and 51 should be supported by some literature sources.

References - considering the fact that the year 2022 is coming to the end, I propose to include at least one relevant article published in this year.

Author Response

Dear Reviewer,

Thank you very much for your time and positive opinion about our paper, and for your valuable comments and constructive suggestions.

We agree with you that continuing the study with the enlarged populations would be desirable to improve the reliability of the results.

According to the Reviewer's suggestion, the Abstract has been supplied with some numerical results of the study.

Objectives: The aim of the present study was to examine and compare the relationship between nutritional status, physical activity (PA) level, concomitant chronic diseases, and quality of life (QoL) in community-dwelling (CD) older people and nursing home (NH) residents.

Material and methods: One hundred NH residents aged 60 years and above and one hundred sex- and age-matched CD older adults were examined. The QoL was examined with the EuroQol-5D questionnaire. Nutritional status was assessed with the Mini Nutritional Assessment questionnaire (MNA), anthropometric measures, and bioimpedance analysis (BIA). The 7-Day Recall Questionnaire and the Stanford Usual Activity Questionnaire were performed to evaluate the PA energy expenditure level (PA-EE) and the health-related behaviours (PA-HRB), respectively.

Results: CD subjects presented a significantly higher self-assessment in the VAS scale in comparison with NH residents (CD: 65.3 ± 19.4 vs. NH 58.2 ± 21.4; p<0.05) but there were no differences within the five dimensions of QoL. In NH patients, the VAS scale was not correlated with any of the variables evaluating the nutritional status and body composition, while in the CD group correlated positively with MNA (rS=0.36; p<0.001), % of FFM (rS=0.22; p< 0.05), body density (rS=0.22; p<0.05) and negatively with % of FM (rS= -0.22; p<0.05). In an institutional environment, only concomitant diseases (mainly urinary incontinence) were found as independent determinants for QoL. In the community, independent determinants of QoL besides concomitant diseases (mainly ischaemic heart disease) were nutritional status or PA-HRB.

Conclusion: Determinants of QoL are different depending on the living environment the older adults. Proper nutritional status and beneficial PA behaviours, are crucial for higher QoL of CD elderly, while for NH residents the main determinants of QoL are chronic conditions.

Text included between lines 42 and 51 has been supported by references number 6 (Bellomo et al.), and 7 (Fiorilli et al.), both published in 2022.

Among the different factors that may be connected with QoL in older people, improper nutritional habits and physical inactivity seem to be the most crucial since they are modifiable and moreover, have become a major problem in the field of public health [6]. The prevalence of malnutrition in older subjects is very high and unfortunately, very often remains unrecognized. Likewise, low PA becomes a major problem due to its influence on functional decline in advanced age [7].

  1. Bellomo, R.G.; Saggini, R.; Barbato, C. Improving “quality of life” through exercise and proper nutrition. J Sports Med Ther. 2022, 7, 010-015.
  2. Fiorilli, G.; Buonsenso, A.; Centorbi, M.; Calcagno, G.; Iuliano, E.; Angiolillo, A.; Ciccotelli, S.; di Cagno, A.; Di Costanzo, A. Long Term Physical Activity Improves Quality of Life Perception, Healthy Nutrition, and Daily Life Management in Elderly: A Randomized Controlled Trial. . Nutrients 2022, 14, doi:10.3390/nu14122527. PMID: 35745256; PMCID: PMC9229916.
